



# Global seamless tidal simulation using a 3D unstructured-grid model

Y. Joseph Zhang[1], Tomas Fernandez-MontBlanc[2], William Pringle[3], Hao-Cheng Yu[1], Linlin Cui[1], Saeed Moghimi[4]

[1]Center for Coastal Resource Management, Virginia Institute of Marine Science, College of William & Mary, Gloucester Point, 23062, USA

[2]Universidad de Cádiz, 11003, Spain

[3]Environmental Science Division, Argonne National Laboratory, Lemont, IL, 60439, USA

[4]NOAA National Ocean Service, Silver Spring, MD, 20910, USA

*Correspondence to*: Y. Joseph Zhang (yjzhang@vims.edu)

**Abstract.** We present a new 3D unstructured-grid global ocean model to study both tidal and non-tidal processes, with a focus on the total water elevation. Unlike existing global ocean models, the new model resolves estuaries and rivers down to ~8m without the need for grid nesting. The model is validated with both satellite and in-situ observations for elevation, temperature and salinity. Tidal elevation solutions have a mean complex RMSE of 4.2 cm for M2 and 5.4 cm for combined 5 other major frequencies in the deep ocean. The non-tidal residual assessed by a tide gauge dataset (GESLA) has a mean RMSE of 7 cm. For the first time ever, we demonstrate the potential for seamless simulation, on a single mesh, from the global ocean into several estuaries along the US west coast. The model is able to accurately capture the total elevation, and qualitatively capture the challenging salinity intrusion processes in the Columbia River. The model can therefore potentially serve as the backbone in a global tide-surge and compound flooding forecasting framework.

## 1 Introduction

Global ocean modelling traditionally focuses on large-scale processes but is increasingly looking into the roles played by smaller-scale processes (internal gravity waves (IGW), topographic and lee waves etc) in the global energy budget (see a review by Arbic et al. 2018, hereafter A18). The state-of-the-art global ocean models now boast 1/12° or finer resolution (thus fully eddying resolving) (A18). More and more models are incorporating barotropic and/or baroclinic tides in their simulation, due to their importance in ocean mixing and global energy budget. Both structured- and unstructured grid (UG) based models have been successfully developed for global tides, starting from the simpler 2D barotropic model, with or without assimilation of altimetry observation. Prominent examples include highly accurate altimetry-informed TPXOv9 (Egbert and Erofeeva 2002) and FES2014 (Lyard et al. 2021) tidal databases. Recently, Pringle et al. (2021) used a 2D model to accurately simulate tide and surge concurrently with high-resolution areas of the mesh focused on hurricane landfall regions. While such a 2D model cannot simulate baroclinic effects, it has been suggested that including a baroclinic term derived from a separate 3D ocean circulation model could be used to improve the energy spectrum of modelled sea surface heights, particularly at the low





frequency end (Pringle et al., 2019). 3D baroclinic models that include concurrent simulation of eddying circulation and tidal motion are also becoming feasible (Arbic et al. 2010; Savage et al. 2017. Wang et al. (2022) recently proposed a reduced-layer (9 layers) 3D baroclinic model with nudged temperature and salinity fields for improving operational total water level forecasts.

As the global ocean constitutes a quasi-closed system, simulating global ocean processes with both tidal and non-tidal frequencies should include coastal oceans where most of the tidal energy is dissipated; collectively, the shelves dissipate about 70-75% of the tidal energy (A18). Also, as A18 mentioned, improving nearshore tides will have a back-effect that will also improve open-ocean tides, especially in tidally energetic areas (e.g., north Atlantic). Accurately accounting for this dissipation process requires high resolution nearshore to represent the complex geometry and bathymetry found therein, which represents

one of the grand challenges in ocean modelling (Holt et al. 2017). In fact, we suspect that the need for several additional types of drags (wave drags etc) in the global ocean modelling in contrast to basin-scale modelling might be related to the inadequate representation of coastal oceans, as even the state-of-the-art resolution of $1/48°$ (Savage et al. 2017) is barely resolving the coastal features. The need for high resolution in the coastal ocean would inevitably strain the already high computational cost associated with global simulations, and in this regard UG models can effectively mitigate the cost. To this day, however, few

UG global models resolve both tidal and non-tidal processes. Logemann et al. (2021) assessed the impact of coastal refinement on tides using an UG global ocean model but did not systematically compare their results with TPXO solution in the deeper ocean, and the reported error metrics appear unsatisfactory.

Despite the tremendous progress made, so far no 3D models for the global ocean exist that can simultaneously include estuaries and rivers without resorting to grid nesting. This is related to the very different characteristics between global and coastal

oceans and estuaries (Fringer et al. 2019); chief among those differences are the drastically different spatial scales and force balances (geostrophic vs ageostrophic; weakly vs strongly forced regimes). The stability and efficiency constraints that come with small-scale processes as commonly found in the coastal/estuarine regimes are formidable. For this reason, some global models have or are developing their own versions of 'coastal model' components that are intended to be nested into the corresponding global models to better close the energy budget (e.g., Andosov et al. 2019).

In this paper, we present a new 3D baroclinic UG model for the global ocean that incorporates both tidal and non-tidal processes and their interactions and is capable of resolving both ocean basins and the estuaries with a *single* mesh. For topics as large as this, inevitably we have to focus on a subset of interests. Here we focus on the short-term predictability (on the scale of 1 year or shorter) of the Total Water Level (TWL) including both tidal and non-tidal components, with the ultimate goal of building a global storm tide and compound flood operational model that simultaneously resolves both eddying motions and some small-

scale processes found near the islands and inside estuaries and rivers of interest. This represents a bold approach of ocean modelling and can effectively close the last remaining gap in the energy budget in the global ocean.

We will first describe in Section 2 the observational datasets used in this paper, as well as the 3D UG model and its setup for the global ocean simulation. We proceed to model validation and assessment of tidal and non-tidal elevation, temperature and salinity in Section 3. In Section 4, we highlight the importance of representation of the ice shelves near Antarctica in the model

bathymetry. Lastly, we demonstrate the model's potential capability in capturing small-scale processes inside estuaries through





a qualitative assessment in the Columbia River; more detailed quantitative assessment would entail site specific calibration and is left for future study, and we also discuss the need for closing the gaps in the theoretical understanding of the cross-scale processes. A short summary is presented in Section 5.

## 2 Method

### 2.1 Observation

In this paper we will primarily validate the model using a satellite-derived reanalysis product OSTIA (for sea surface temperature (SST)), an altimetry-informed global tidal model TPXOv9, a global tide gauge dataset (GESLA) for sea-surface height (SSH), and ARGO floats for temperature and salinity profiles. Tide gauges observation will be used sparingly as most of these are located in complex nearshore regions that require accurate bathymetry information and more mesh work to capture that information. We will use a few tide gauge data for a target study of the US west coast in Section 4. Our ultimate goal is to build a global 3D UG model that is able to seamlessly transition from ocean basin into small creeks; to achieve this goal, however, it's essential that the model has sufficient skills for large-scale processes first. We note that the current 3D UG model has been extensively applied and validated in many coastal regions so the latter does not present fundamental challenges for the model as long as a proper calibration procedure is followed.

### 2.2 Model description

SCHISM (schism.wiki) is an open-source community model solving 3D hydrostatic form of the Navier-Stokes equation with Boussinesq approximation (Zhang et al. 2016). Major innovative features of SCHISM include: (1) semi-implicit time stepping scheme that bypasses the most stringent stability constraints (and thus allows very fine resolution of O(1m) without the need to reduce the time step); (2) a highly flexible 3D gridding system, with hybrid quadrangular-triangular unstructured mesh in the horizontal dimension and localized sigma coordinates with shaved cells (LSC$^2$) in the vertical dimension (Zhang et al. 2015). The flexible gridding system enables powerful 'polymorphism' with a single SCHISM grid being able to seamlessly morph between full 3D, 2DH, 2DV and qusi-1D configurations (Zhang et al. 2016); (3) judicious combination of higher- and lower-order schemes to ensure accurate representation of diversity of processes from creek to ocean basin scales (Zhang et al. 2016; Ye et al. 2019). These features has allowed a single model to be used for challenging compound flooding studies that involve coastal transition zones between hydrodynamic and hydrologic regimes, forced by ocean, precipitation and watershed rivers (Ye et al. 2020; Zhang et al. 2020; Huang et al. 2021; Ye et al. 2021).

The global unstructured mesh in the horizontal dimension consists of ~4.6 million nodes, ~9 million triangular elements, with nominal resolution of 10-15km in the ocean basin (Figs. 1-2), thus barely eddy resolving. The mesh resolution generally increases to ~3km at most of the coastline of the continents or islands; higher resolution of ~1-2km is applied in North America and western Pacific due to our interests in those regions. As an illustration, we have also added detailed representations of a few estuaries and rivers along the US west coast (Fig. 1(b-e)). We demonstrate in Section 4.2 the model's potential for seamless





cross-scale transition into nearshore and estuaries, with sufficient resolution to resolve salt intrusion process and a minimum element size of ~8m found near a coastal highway inside the Columbia River estuary (Fig. 1e). Therefore, the mesh size spans 4 orders of contrast (from ~10km to ~8m). Overall, about 50% of elements have resolution 5km or higher (Fig. 2c).

Consistent with our main goal in this paper, we use Gebco (GEBCO Compilation Group 2019) as the main DEM sources, with a resolution of 500m for global oceans. This resolution is adequate in most of the coastal and deep oceans but is not sufficient for nearshore and estuaries. As shown in Section 4.1, it's important to include the ice shelf effect on the bathymetry in the Southern Ocean, and we therefore use RTOPO for this purpose (Schaffer et al. 2016). To improve the model skill in the target estuaries in the US west coast, we have locally utilized a hierarchy of DEMs from NOAA's Coastal Relief Model (~90m

resolution) and CUDEMs (1-10m resolution; CUDEM 2022) and USGS's CoNED (1-3m resolution; CoNED 2022).

The 3D model takes full advantage of the flexible vertical gridding system (LSC$^2$; Zhang et al. 2015). The number of sigma layers varies from a maximum of 34 to 1 (i.e. 2DH configuration), with an average of 32 layers. Using 1 layer in shallow and dry areas greatly improved the efficiency and robustness of the model (Huang et al. 2021). The vertical high resolution is focused on the near-surface zone at the expense of the bottom in order to conserve computational cost. As a result, the near-

bottom vertical layers can be as thick as 1km in the deep ocean; in other words, the logarithmic layer there is not well resolved and therefore, we apply zero friction in the deep depths. To ensure adequate energy dissipation toward shallows, we use a simple depth-dependent bottom friction coefficient (used in the quadratic drag formulation) that linearly increases from 0 at depth 200m to 0.0025 at 50m. For the sake of simplicity, no attempt has been made to optimize the friction in each region yet, and this is left for future work.

A main advantage of using a 3D barolinic model is that it accounts for internal tides (ITs), whose production over open-ocean topographic features accounts for about 25-30% of the energy lost in the global barotropic tidal energy budget (A18). The results from Egbert and Ray (2003) suggest IT dissipation is an important contributor to the mixing that underpins the large-scale circulation (Munk and Wunsch 1998). According to A18, a horizontal resolution of at least 1/10° is needed for a fully vigorous low-mode IT field. On the other hand, a horizontal resolution of about 1/24° or finer is necessary for simulating a

vigorous IGWs continuum. Therefore, the current model, because of the limited computational resources available to us, only resolves ITs but not lGWs. Although parameterized IGW drag formulation (e.g., Garner 2005) could be included in the model, its inclusion in a baroclinic model is tricky as part of the signal is already resolved (A18). Therefore, we neglect IGW drag here.

The semi-implicit model uses a non-split time step of 120s, a turbulence closure scheme of *k-kl* of the Generic Length Scale

Model (Umlauf and Burchard 2003), and a bi-harmonic viscosity (Zhang et al. 2016). Since no bathymetry smoothing was done in our mesh, the presence of very steep bottom slopes near numerous islands and ocean trenches requires additional momentum stabilization than provided by the bi-harmonic viscosity. Therefore, we add a Smagorinsky-type viscosity that is designed to 'penalize' the steep slopes. In line with the Shapiro filter-like implementation of viscosity inside SCHISM that works well with highly distorted UGs (Zhang et al. 2016), the Smagorinsky-type viscosity is implemented as:





$\gamma=0.5\ \tanh(C\Delta t\Gamma)$                                                                                  (1)

$$\Gamma = \left[ u_x^2 + v_y^2 + \frac{(u_y^{\square} + v_x^{\square})^2}{2} \right]^{1/2},$$

                                                                                       (2)

where $\gamma$ is the filter strength with a maximum value of 0.5 (Zhang et al. 2016), $\Delta t$ is the time step, $\Gamma$ is the deformation rate, and $C$ is a non-dimensional constant specified by the user. In the global model, we found that $C$=1000 is sufficient to suppress most spurious noise in the horizontal velocity field. For tracer transport, the 3rd-order WENO scheme (Ye et al. 2019) is used

at depths greater than 10m (whereas an upwind scheme is used for shallower depths).

The primary validation period used in this paper is a 120-day window from June 1, 2011 to September 29, 2011. Atmospheric forcing from ERA5 (with a resolution of 25km) was applied onto the ocean surface, including wind, air pressure, precipitation, and heat fluxes. Relaxation of temperature and salinity near the ocean surface, which is commonly utilized in many global ocean models (Ringler et al. 2013), was not applied here due to the relatively short duration of the simulation. The tidal

potential with 6 constituents (M2,S2,N2,K1,O1,Q1) and self-attracting and loading (SAL) tides were included using the depth-dependent parameterization of Stepanov and Hughes (2004). For the harmonic analysis and comparison with TPXOv9, we used the model results from Days 20 to 60 and turned off the atmospheric forcing, as the 40-day results used were long enough to distinguish the 6 constituents used in the analysis. The analysis was focused on the main tidal constituent of M2 as in other studies (Pringle et al. 2021), since it accounts for most of the tidal energy.

The model was initialized with a dynamic flow field interpolated from the global 1/12° (~10km) HYCOM. Linear interpolation was used in the interpolation of sea-surface height (SSH), horizontal velocity, salinity, and temperature. Constant extrapolation was used in regions not covered by HYCOM. The HYCOM derived SSH and horizontal velocity represent the non-tidal component; once started, the tidal potential and SAL will initiate the tidal motion in the system. Altogether 868 largest rivers were included along the coast, with monthly mean flow information from Dai (2021). The river temperature input was set to

be the 'ambient' flow temperature due to the lack of such information. Starting the simulation from a fully dynamically equilibrated flow field allows us to significantly reduce the time required for warming up the model. Still, the discrepancies between HYCOM and SCHISM and initiation of the tides require a short ramp-up period, estimated to be shorter than 20 days. We remark here on a few limitations of the current model. The need for IGW drag may be reassessed in the future. Alternative implementations of SAL, e.g., interpolation from TPXO or FES should be explored. In the high latitudes, the lack of an ice

model is a major gap for the model; even though we have this component inside the SCHISM system, its inclusion would significantly increase the computational cost especially under high mesh resolution. Lastly, atmospheric forcing could use higher spatial and temporal resolution for more accurate simulation of coastal processes especially during storm events.

SCHISM's good parallel scaling allows efficient simulation given adequate resources. For example, on 5600 cores of an Intel cluster (TACC's Frontera) the model is able to run ~120 times faster than real time. Doubling the core count to 11200 increases

this real-time ratio to ~223, which translates to an excellent parallel efficiency of ~95%.





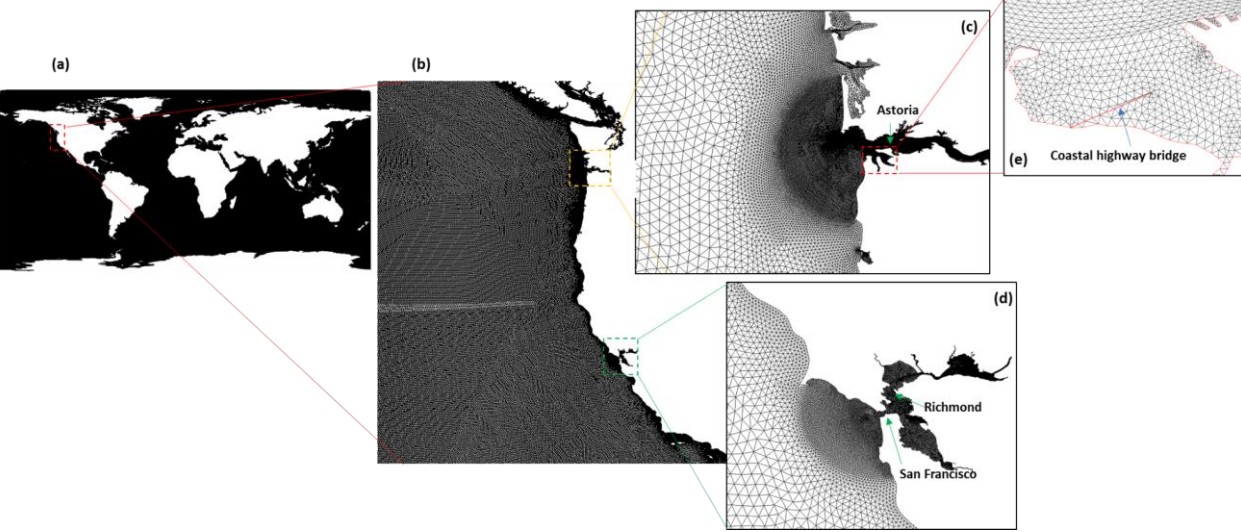

**Fig. 1: (a) Global unstructured mesh, with (b) successive local refinements in the US west coast, including (c,e) the Columbia River and estuary, and (d) San Francisco Bay.**

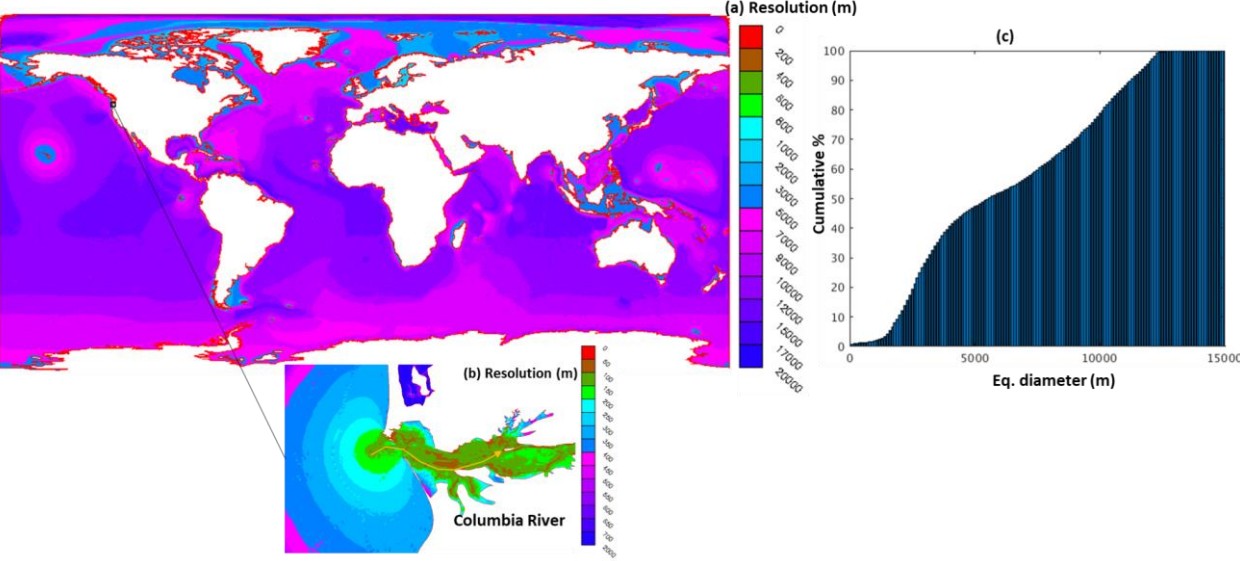

**Fig. 2: (a) Mesh resolution as measured by equivalent diameter, with (b) zoom-in near the Columbia River; the orange transect will be used to show salt intrusion path. (c) Histogram of resolution.**





## 3 Model validation

We start the model validation with surface elevation, with respect to both tidal (Section 3.1) and non-tidal components (Sections 3.2). For 3D variables, we focus on temperature and salinity as these are the major drivers of the large-scale processes due to their contributions to the uneven oceanic mass distribution. Note that the focus of this study is on the accurate prediction of the total elevation; for accurate simulation of the eddying processes, finer mesh resolution would be required. We follow HYCOM in using NGVD29 (geopotential) as the vertical datum, which is problematic in the assessment of either the altimetry

(which is referred to a fixed geoid (Jahanmard et al. 2021)) or some tide gauges that refer to different datums (e.g. NAVD88). More rigorous assessment of the total water level is left out for future studies after a geoid-based datum becomes available, and here we focus on the tidal elevation as well as the *variability* of the non-tidal elevation (i.e., the mean biases are removed when datums do not match or are unknown). Standard error metrics are reported below, including: RMSE (Root-Mean-Square Error), MAE (Mean Absolute Error), correlation coefficient and Wilmot score (Wilmot 1981).

### 3.1 Co-tidal chart for M2

Globally, the M2 amphidromes correspond to the local minimum of the tidal energy in the open ocean or near islands (e.g., Taiwan, New Zealand, Madagascar etc) where the tides tend to rotate in the form of Kelvin waves (Fig. 3b). The amplitude maxima, on the other hand, are typically found near semi-enclosed basins near resonant modes (e.g., European Seas, Hudson Bay, Bay of Fundy etc), where the tidal transformation is rather complex (Fig. 3b).

A major benefit of the 3D model is that the time varying dissipation of tidal energy due to internal tides is accounted for inside the model. Therefore, with a simple specification of bottom friction (as a function of depths), the simulated M2 distribution already has a good skill as compared to the benchmark TPOXv9 tidal database (Fig. 3).

The complex RMSE for a constituent is defined as (Wang et al. 2012):

$$RMSE_j = \sqrt{\frac{1}{2}VD_j^2} \quad,$$

(3)

$$VD_j^2 = \left[(A_{\mathrm{o}})_j\cos(\varphi_{\mathrm{o}})_j - (A_{\mathrm{m}})_j\cos(\varphi_{\mathrm{m}})_j\right]^2 + \left[(A_{\mathrm{o}})_j\sin(\varphi_{\mathrm{o}})_j - (A_{\mathrm{m}})_j\sin(\varphi_{\mathrm{m}})_j\right]^2 \quad,$$

(4)

where *VD* stands for vector difference, *A* is the amplitude, 'm' and 'o' refer to model and observation, $\varphi$ is the phase, and *j* is the frequency index. The integrated RMSE for a specific area $\Omega$ is then defined as:

$$\overline{RMSE_j} = \sqrt{\frac{\int_A RMSE_j^2 \, d\Omega}{\int_A d\Omega}} \quad,$$

(5)





The $\overline{RMSE}$ can be computed for a single frequency (e.g. M2) or summed up for a group of frequencies (e.g., the other 5
frequencies than M2) to give a single number for those frequencies.

The averaged complex RMSE for M2 is 4.2cm for depths greater than 1km, and 14.3cm for shallower depths. The averaged
RMSE for the remaining frequencies (S2, N2, K1, O1, Q1) is 5.4cm / 16.6cm or depths greater/less than 1km. These results
are slightly better than the previous best 3D model results without data assimilation (Schindelegger et al. 2018) but worse than
those in Pringle et al. (2021). Note that there are differences in the shallow areas included in each model, which makes the
numbers for the shallows less reliable than those for the deeper depths. Our results are quite satisfactory given the fact that
minimal calibration (with respect to bottom friction etc) was conducted; the more complete physics as incorporated in the 3D
baroclinic model reduced the amount of calibration required to achieve good tidal results compared to 2D barotropic models
(e.g., Blakely et al., 2022). Compared to other global 3D models, our model seems to be able to obtain satisfactory results
without the need for some elaborate drag formulations described in A18, which might be attributed to the fact that the higher
resolution used in the coastal ocean has provided adequate energy dissipation.

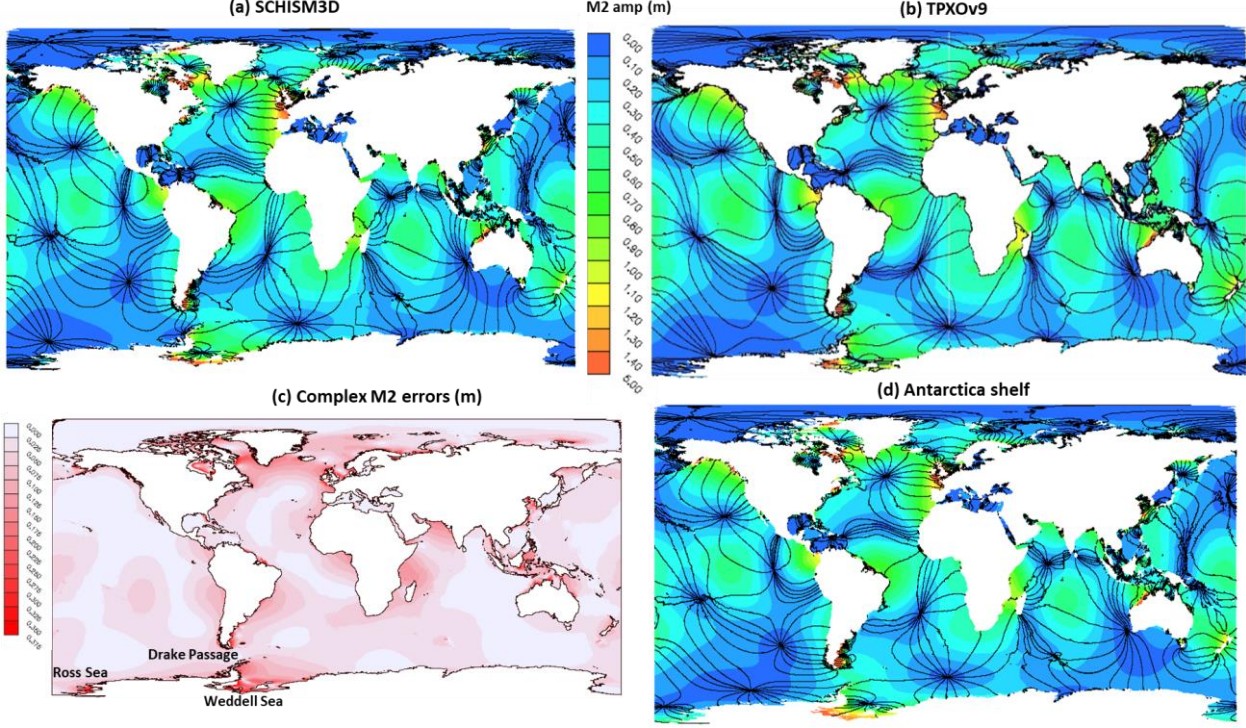

**Fig. 3: comparison of co-tidal chart for M2 between (a) SCHISM and (b) TPXOv9. The complex M2 RMSE is shown in (c); some
geographic locations are labelled here also. (d) Sensitivity results without Antarctica ice shelf represented.**





## 3.2 GESLA tide gauges comparison

The modelled sea levels are compared with observed sea levels from tide gauge stations in the Global Extreme Sea Level Analysis (GESLA) dataset (Woodworth et al., 2016). The tidal harmonic analysis is performed using the t-tide package (Pawlowicz et al. 2002). The skills of the model to reproduce tides are assessed using the RMSE for each tidal component and

the root sum of square (RSS) as an overall skill index:

$$RSS = \sqrt{\sum_{j=1}^{n} RMSE_j^2}$$

(6)

where $n$=6 is the total number of frequencies. The tidal skill scores are also computed from the FES2012 model, an altimetry-informed model used as a reference to evaluate our model.

The assessment of model performance to reproduce the non-tidal residual (NTR) sea level variation is also conducted. Due to

the uncertainty in the vertical datums used in many gauges, NTR time series are obtained by de-meaning for the common period, and de-tiding using the t-tide package. Afterwards skill scores of RMSE and Pearson correlation coefficient are computed. In order to assess the model predictive skill for extreme water levels, RMSE and correlation scores are also computed for the upper tail of the time series, i.e. values exceeding 95th percentile of the observed NTR.

Overall, a satisfactory model performance is observed in coastal areas in comparison with GESLA. The model RMSEs are

less than 0.1 m at ~45% of tidal stations for M2 and less than 0.05 m at ~58% of the stations for S2 (Fig. 4). The comparison with FES2012 indicates larger error in SCHISM (+6cm in M2 RMSE and +2cm for S2 RMSE) (Table 1). It is an acceptable performance given the fact that FES2012 incorporates data assimilation from altimetry data (Carrère et al., 2013). Most of the larger errors occur in areas with DEMs of large uncertainty (e.g., Canadian coasts with fjords; southern Chilean/Argentine coasts) or in areas with insufficient resolution (e.g., European Seas) (Fig. 4). The uneven error distribution may guide future

priority in mesh development.

The model also accurately reproduces the NTR in coastal waters (Fig. 5). The average RMSE is only 7cm with a median value of 6cm; ~88% of RMSEs are below 10cm and only 1.6% exceed 15cm, mostly at stations located in the North Sea and Northwestern Pacific Ocean. The RMSE increases for extreme conditions (NTR>95th percentile), but the model still adequately reproduces the extreme conditions with an averaged RMSE of 11cm and a median of 9cm. In addition, RMSEs are

less than 15cm at ~80% of the tidal stations under extreme conditions. These results are consistent with our previous results using the 3D model (Ye et al. 2020; Huang et al. 2021, 2022), and indeed, the better skill at capturing the NTR is a major advantage of 3D over 2D models, because the NTR is largely driven by the eddying motions and large-scale ocean current systems that originate from the uneven ocean mass distribution.






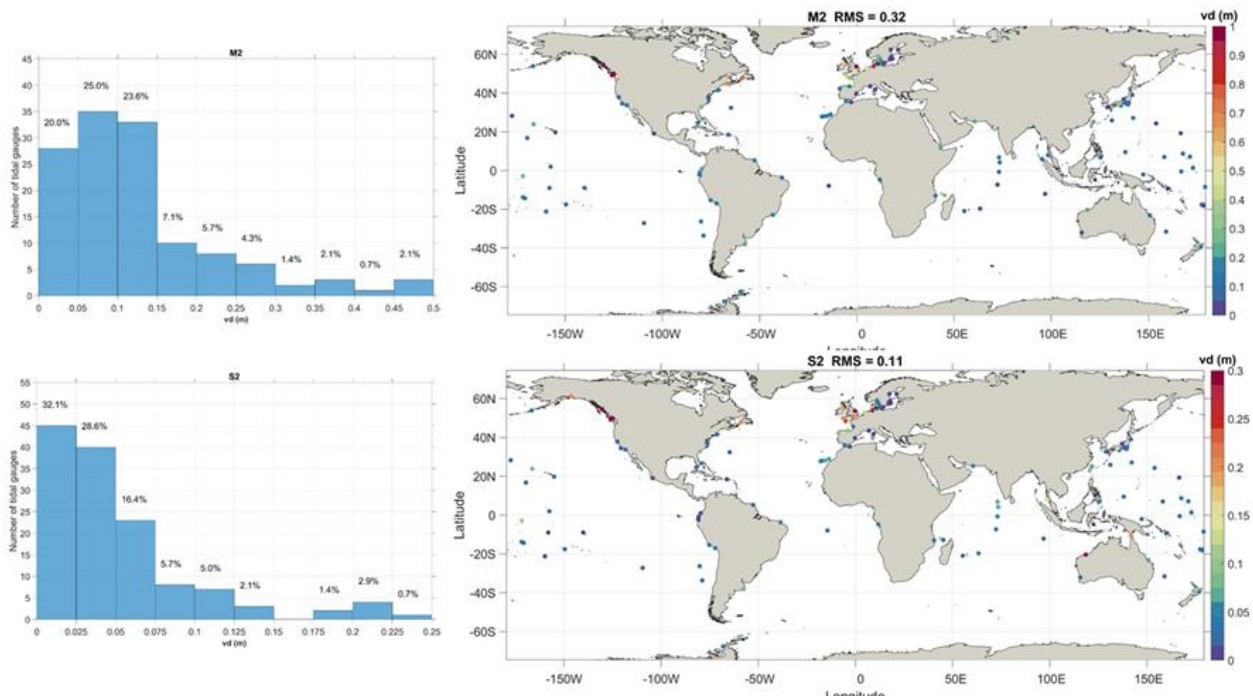

Fig. 4: Histogram and maps of scatter plots of M2 and S2 vector difference error.

Table 1. Summary of model performance to reproduce the main semidiurnal tidal component for SCHISM and FES2012 models against GESLA

|  | RMSE M2 | RMSE S2 | RSS |
|---|---|---|---|
| **SCHISM** | 0.32 | 0.11 | 0.35 |
| **FES2012** | 0.26 | 0.09 | 0.28 |


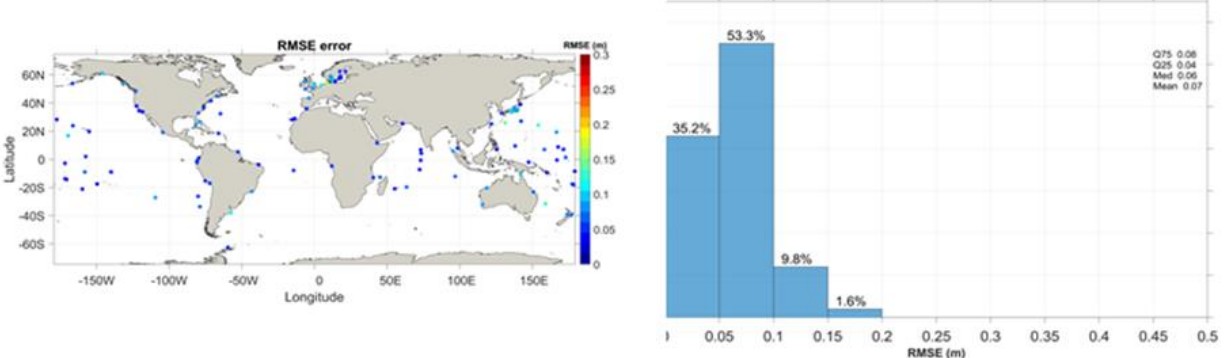

Fig. 5. Maps and histogram of RMSEs for non-tidal residual (NTR). ~88% of RMSEs are below 10cm.



### 3.3 SST

The quality of the simulated SST is assessed against a reanalysis product, OSTIA (Good et al. 2020), which blends different satellites with in-situ data into regular 0.05° resolution sea surface temperature estimation on a daily basis. This dataset provides global SST with an overall analysis error of ~0.4°C. Most large errors are distributed in the Arctic region which can reach 3.6°C maximum. In order to isolate the effects from the Arctic, the analysis is done for the global domain with and without the Arctic region (latitude > 60°N). The OSTIA analysis error excluding the Arctic is reduced to 0.3°C on average.

The simulated SST is mostly similar to OSTIA (Fig. 6). In particular, the model is able to capture major boundary currents (Kuroshio, Gulf Stream etc) as well as equatorial instabilities. A closer look at the comparison reveals larger warm biases in the Arctic region (Fig. 7), which is not surprising because we did not include the ice component in our model. Excluding the Arctic the averaged MAE stays much lower at ~0.8 °C throughout the simulation period (Fig. 7b). Besides the high latitude regions in the northern hemisphere, relatively higher biases are also found near the boundary currents and equatorial instability

regions (Fig. 7a).

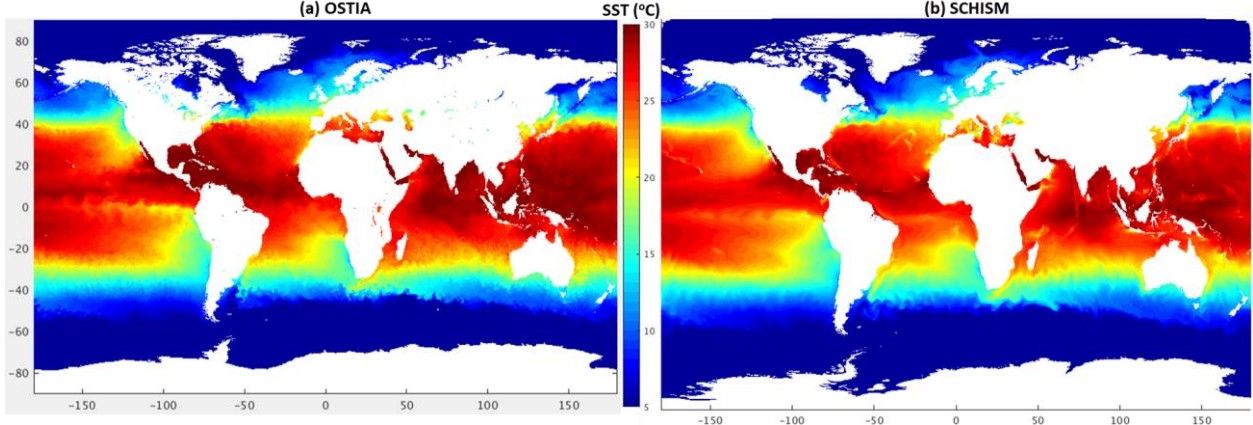

**Fig. 6: Comparison of SST between (a) OSTIA; (b) SCHISM at the end of 120-day simulation (Sept 29, 2011).**





**Fig. 7: SST from the 120-day run from June 1, 2011. (a) SST Bias; (b) MAE.**

**3.4 SSS**

There is a lack of salinity observation at large scales. Although NASA's Aquarius satellite missions did cover the simulation period, the data is not publicly available yet and, in any case, may be too coarse for our purpose. Therefore, we use HYCOM (which has assimilated profile data) as a reference solution in our comparison. The two models are largely comparable, including major fronts and instabilities near equator, the freshwater plumes and boundary currents, and intrusion from north

Atlantic into Arctic etc (Fig. 8). Freshwater plumes from Amazon and other large rivers appear to be larger in SCHISM. Due





to the absence of an ice component in our model, there are also some differences in the Arctic Ocean. Overall, the modelled SSS is satisfactory.

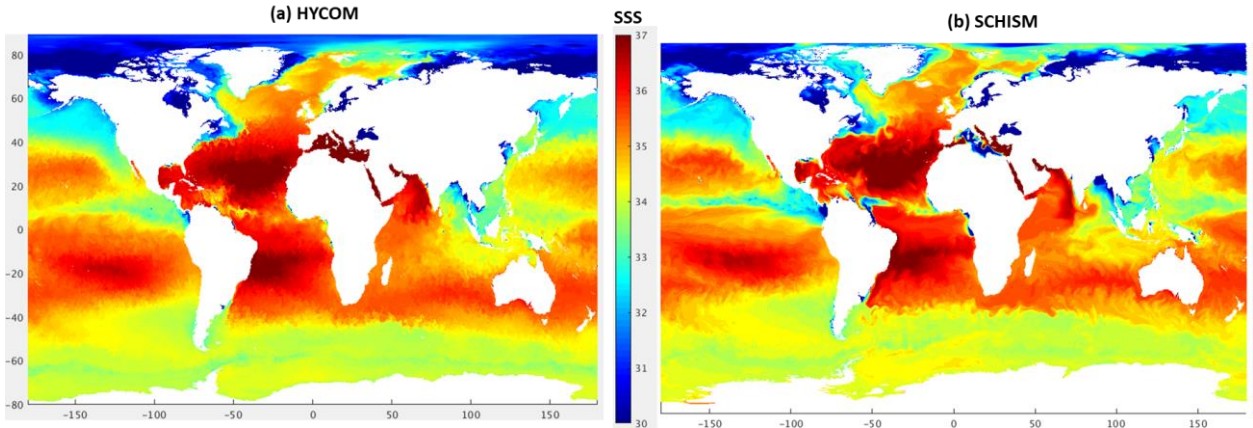

**Fig. 8: Comparison of SSS between (a) 1/12° HYCOM and (b) SCHISM at the end of 120-day simulation (Sept 29, 2011).**

## 275  3.5 ARGO profiles

To assess the model skill in capturing the vertical structures of temperature and salinity, we use all ARGO data in each ocean basins except the Arctic (Fig. 9); as seen in the previous sections, the model has larger biases in the Arctic due to the absence of an ice component.

Total number of ARGO profiles is around 330 average per day in our simulation period. The model results are first interpolated
from surface to 2000m to match the measuring range of ARGO. We follow the original ARGO data structure to divide our analysis into 3 different basins (Atlantic, Pacific and Indian Oceans), each including parts of the Southern Ocean. Due to the relatively small number of profiles (less than 30 per day) available near the surface (< 6m) in all basins, 0-6m data are excluded to produce more reliable statistics.

Overall, the modelled temperature and salinity profiles are satisfactory, with a MAE of ~0.6°C for temperature and ~0.2PSU
for salinity (Fig. 10). Of all ocean basins, the Pacific has the smallest biases (Fig. 11). The simulated temperature in all basins tends to have a cold bias (~-0.4°C) near the surface, and biases below 200m depth are smaller. The simulated salinity in all basins has a positive bias below 1400m depth (Fig. 11), which is inherited from the initial condition. The good skills for temperature and salinity profiles are also confirmed by the high correlation scores that exceed 0.9 most of the time (Fig. 12).



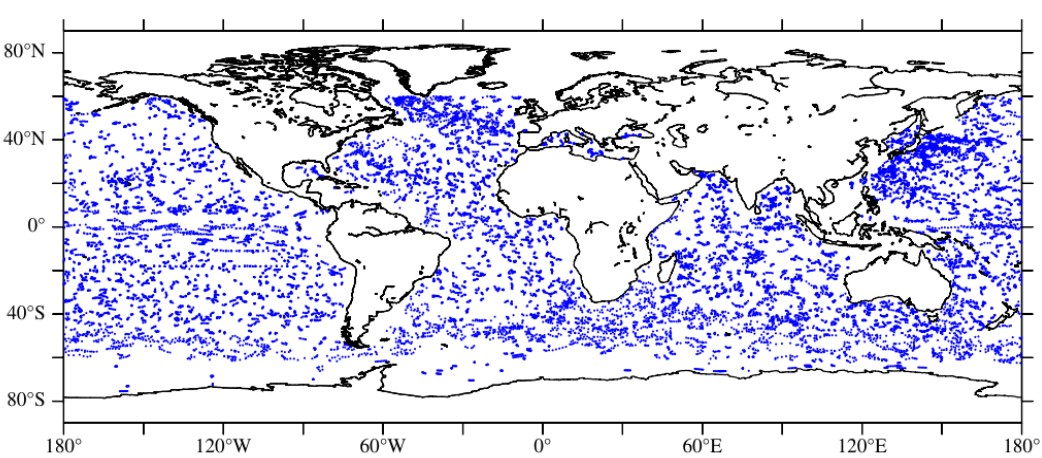

**Fig. 9: Locations of ARGO profiles used in this study (Period: Aug/1 ~ Sep/28 2011, 20823 profiles in total)**

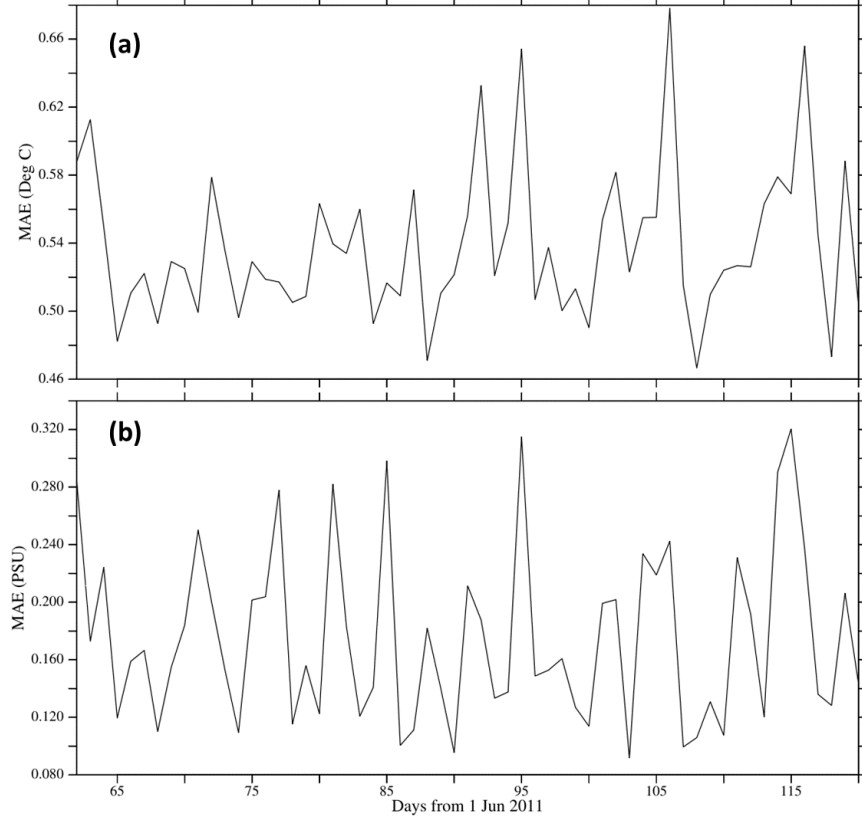

**Fig. 10: Averaged daily MAEs from all ARGO profiles for (a) temperature and (b) salinity.**





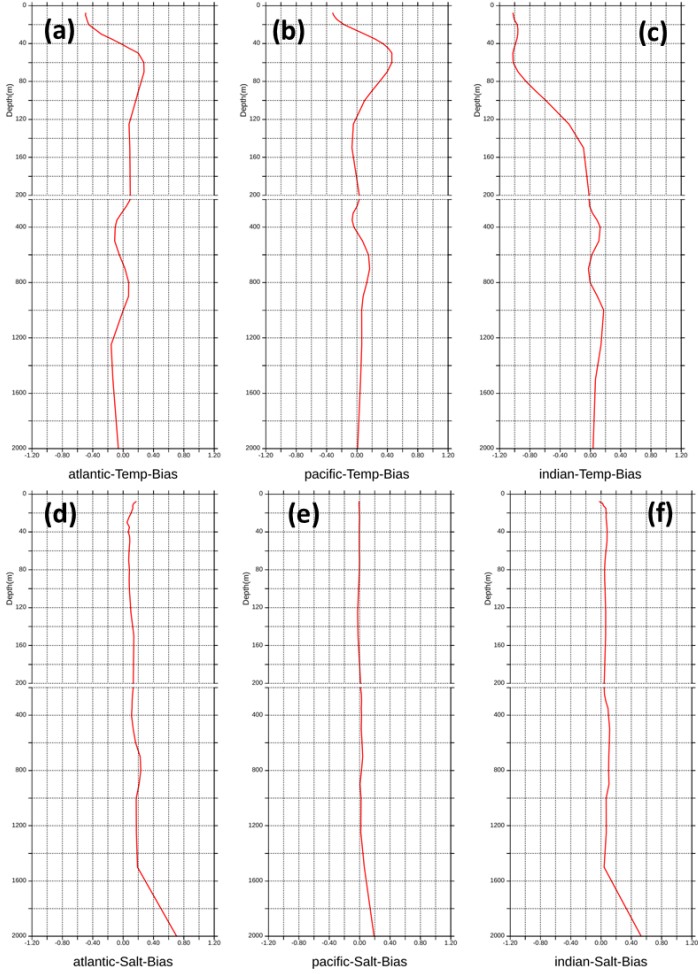

**Fig. 11: Mean biases in each ocean basin. (a-c): temperature biases in Atlantic, Pacific and Indian Oceans respectively; (d-f): salinity biases. Each plot is divided into two parts to clearly show the larger biases in the top 200m depth.**

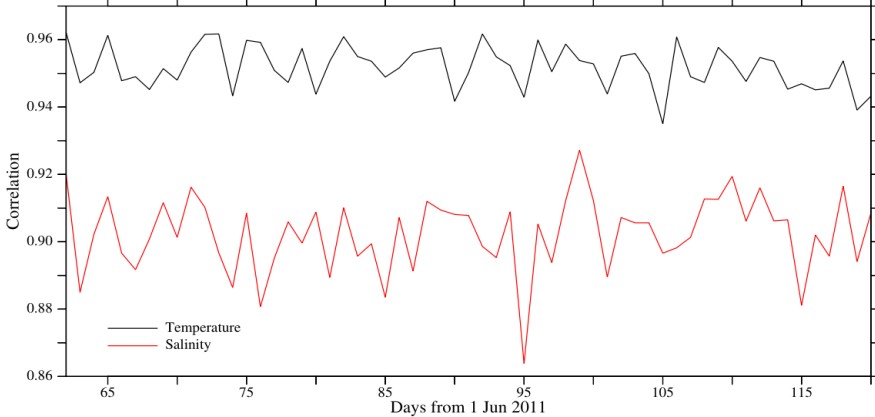

**Fig. 12: Averaged daily correlation coefficients for all ARGO profiles.**



## 4. Sensitivity tests and discussion

Calibrating a 3D baroclinic model like ours can be an expensive exercise, especially for 3D variables such as temperature, salinity and velocity. However, since the focus of this paper is on the water surface elevation, we found through various sensitivity tests that the elevation results are sensitive to only a few parameters, including the bathymetry and bottom friction, because the thermo-steric contribution to elevation has been accounted for in the model. Compared to 2D models, the more complete physics embedded inside 3D models greatly simplifies the calibration process, e.g. the time-varying IT induced

dissipation is already included in the model. This finding is consistent with our previous finding for a sub-domain of the US east coast and Gulf of Mexico coast (Huang et al. 2022). In this section we will show the model sensitivity to the representation of the ice shelf effects in the Southern Ocean.

    A main novelty of our 3D UG model is its capability to seamlessly traverse scales from global oceans into very localized scales as found in the estuaries and rivers. To the best of our knowledge, this capability has not been demonstrated before without

resorting to grid nesting. Therefore, our model represents a major advancement in efficiently simulating global and local scales in a *single* UG model, thus allowing the interaction and connection among scales to be fully examined in our model. We demonstrate this potential here using two estuaries in the US west coast as example.

### 4.1 Southern Ocean: ice shelf effect

    Bathymetry is known to play a pivotal role in the tidal and non-tidal processes (Ye et al. 2018; Huang et al. 2022). As far as

the global tide is concerned, one particularly important region is the Southern Ocean, which has an extensive distribution of ice shelves along the Antarctica coast. The existence of ice shelves effectively changes the local bathymetry, which affects tidal propagation locally and beyond (Blakely et al., 2022). Without accounting for those shelves, an erroneous amphidrome appears between Drake Passage and Ross Sea (Fig. 3d vs 3a), and the amphidrome just east of the Ross Sea (Fig. 3a) is displaced westward (Fig. 3d). Other differences are also visible south of Australia and in the Weddell Sea (Fig. 3d vs 3a).

### 4.2 Into estuaries

    As a proof of concept, we illustrate the model's potential in traversing from large oceanic to small estuarine scales using a few estuaries in the US west coast as an example but leave detailed calibration and validation to future studies. Obviously, the calibration process will be expensive given the large mesh size used here. However, we show that with local mesh refinement and minimal calibration done in the current 3D model, the model is able to qualitatively capture some small-scale processes.

The Columbia River and San Francisco Bay (SFB) are the two largest estuaries in the US west coast (excluding Alaska), and characterized as meso-tidal ssytems. The river discharges vary greatly between the two systems and over time. The Columbia River has a long-term mean flow of $2.5–11 \times 10^3$ m$^3$/s over a typical year (Bottom et al., 2005). SFB receives most of the freshwater inputs from the north Bay which is connected to the Sacramento-San Joaquin Delta, and the net Delta outflow is smaller than the Columbia River (from ~500m$^3$/s to 2000m$^3$/s). Both systems exhibit similar seasonality in the river flow, with



the lowest flow occurring in late summer and highest flow during spring freshets. Due to the specificity of the forcings experienced by the two systems, the salt intrusion processes are quite different. The Columbia River estuary shows a strong spring-neap variation in the stratification, and occasionally exhibits salt edge conditions (Jay and Smith 1990). In the MacCready-Geyer estuarine classification diagram, the Columbia River Estuary (CORIE) is classified as a 'time dependent salt wedge' system (Geyer and MacCready 2004). The shorter shelf width near the CORIE makes it more susceptible to the

prevailing coastal upwelling that is more common along the Oregon-Washington shelf. Previous modelling studies (Karna and Baptista 2016) indicate that the Columbia River processes in particular are extremely challenging for numerical models. The SFB, on the other hand, represents a typical partially mixed estuary (Geyer and MacCready 2004).

The total elevations at two NOAA tide gauges in SFB and one gauge in CORIE are assessed in Fig. 13. Fortuitously, the vertical datum used in the model, NGVD29 (inherited from HYCOM) is close to the Local MSL used at these 3 gauges, and

therefore, no adjustment of the vertical datum is necessary. For other coastal regions (e.g., US east coast), the datum differences can be substantial, which calls for a geoid-based datum for regional and global tidal models (Jahanmard et al. 2021). With particular attention paid to the mesh representation near these two estuaries, the model is able to accurately capture the tidal and non-tidal elevations in the two systems, as shown in Fig. 13, with RMSEs for the total elevations between 8.3cm (at Richmond) and 18.7cm (at Astoria), and the correlation coefficients and Wilmot scores all exceeding 0.98. As mentioned in

Section 3.2, one major advantage of 3D models over 2D models is their ability to better capture the NTR and thus TWL. Our experience suggests that with similar mesh refinement procedure, reliable DEMs and some calibration with respect to the bottom drag coefficients, similar elevation skills can be obtained for other estuarine systems.

The Columbia River plume is a major coastal feature in the region and can extend 100s of kilometres offshore (Baptista et al. 2005) and has a major impact on the ecosystem (Burla et al. 2010). The plume is highly dynamic and mostly wind driven but

modulated by tides and river discharge (Burla et al. 2010). Fig. 14 shows a 'canonical' view of the plume when the wind forcing is weak or relaxed. The combination of the Coriolis and inertial forces turns the plume northward with a coastally trapped jet in the north (in the form of Kelvin waves) and visible recirculation inside the freshwater bulge (Garcia-Berdeal et al. 2002).

The salt intrusion into the Columbia River is illustrated just before a spring freshet during a spring tide (Fig. 15). The intrusion

path largely follows the two main channels: the natural and shorter north channel and the dredged shipping channel in the south. The intrusion extent is qualitatively consistent with the previous results (Zhang and Baptista 2008); in particular, the maximum intrusion during the flood tide reaches the upper estuary along the south channel whereas the minimum intrusion during the ebb tide is near Astoria, with several low-lying tidal flats exposed (Fig. 15).

As implied in Figs. 14&15, the salt intrusion can lead to a highly stratified estuary. The large stratification is due to a

combination of high river discharge, channelized and dredged (for navigation) bathymetry, and tidal asymmetry (Baptista 2006). The strongly stratified intrusion pattern is illustrated along the dredged south channel near the maximum flood and ebb in a neap tide (Fig. 16). A sharp interface can be clearly seen especially during the ebb phase (Fig. 16b). It is well known that the numerical (spurious) dissipation needs to be carefully controlled in the numerical schemes in order to capture the very




sharp and wedge-like stratification as in the Columbia River. The combination of the 3D gridding system, 3rd-order WENO

transport scheme (Ye et al. 2019), the horizontal and vertical mixing schemes (Zhang et al. 2016) in SCHISM proves adequate

for this challenge.

Although a more careful validation against observation (including the vertical salinity profiles) is necessary to ascertain the

model skill in capturing this type of smaller-scale 3D processes, which would inevitably involve site-specific parameterization

and calibration procedure, the preliminary results shown here are very promising and offer the potential to finally close the

gap in simulating the global ocean-estuary-river-lake continuum. Note that the model does allow specification of spatially

variable parameterizations such as bottom drag and horizontal mixing scheme, which will be necessary in future calibration

process. On the other hand, the physical justifications of these choices, in the context of cross-scale processes from global

ocean to estuaries and rivers, warrant further research, e.g., traditionally different horizontal mixing schemes have been used

in the two regimes (Fringer et al. 2019).


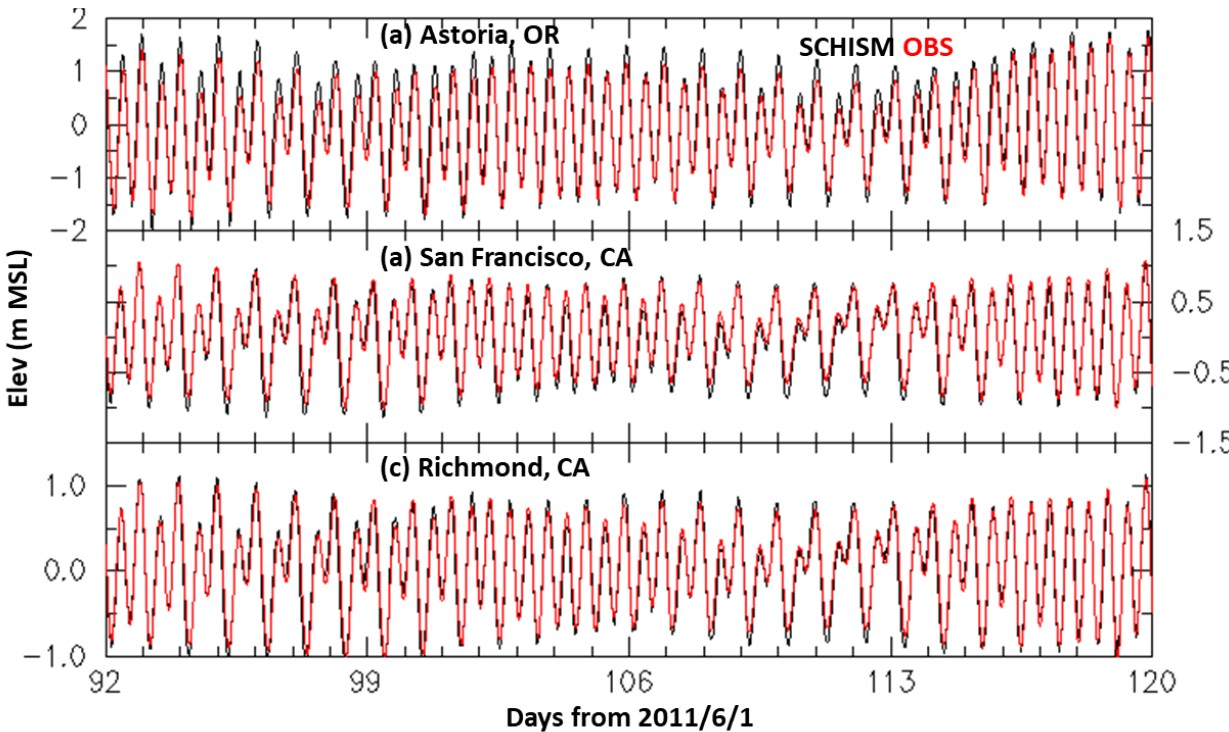

**Fig. 13: Comparison of elevation at 3 tide gauges in Columbia River and SF Bay, USA. See Fig. 1(cd) for gauge location. The RMSEs are 18.7, 9.4 and 8.3 cm, Correlation Coefficients are 0.98, 0.98 and 0.98, and Wilmot skills are 0.98,0.99 and 0.99, respectively. Both model and data use NGVD29 as the vertical datum.**

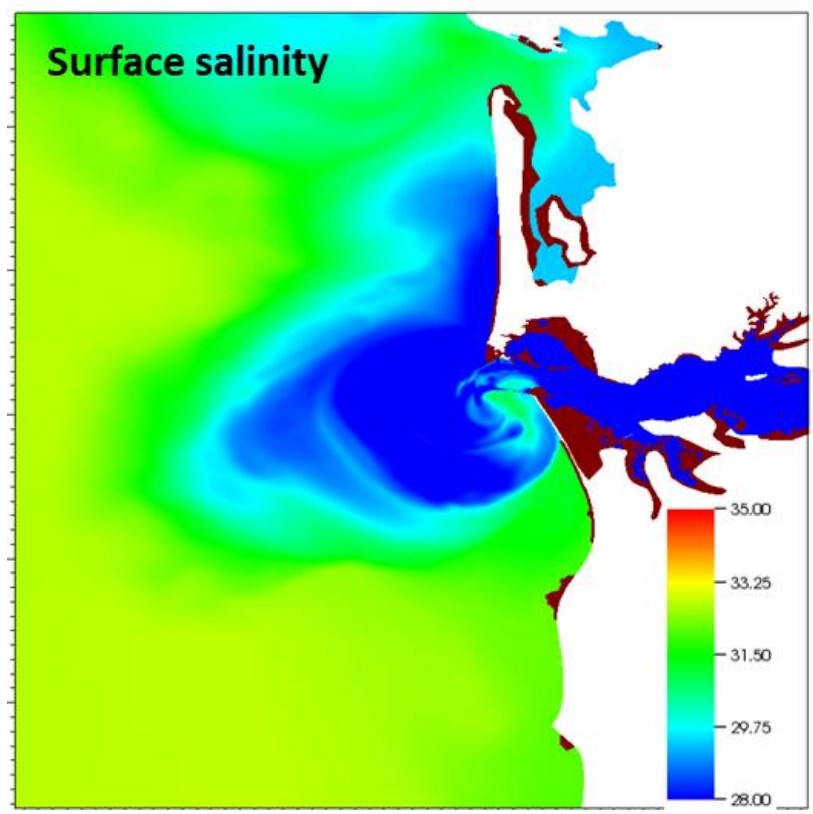


**Fig. 14: Surface salinity plume near the Columbia River during a wind relaxation period.**

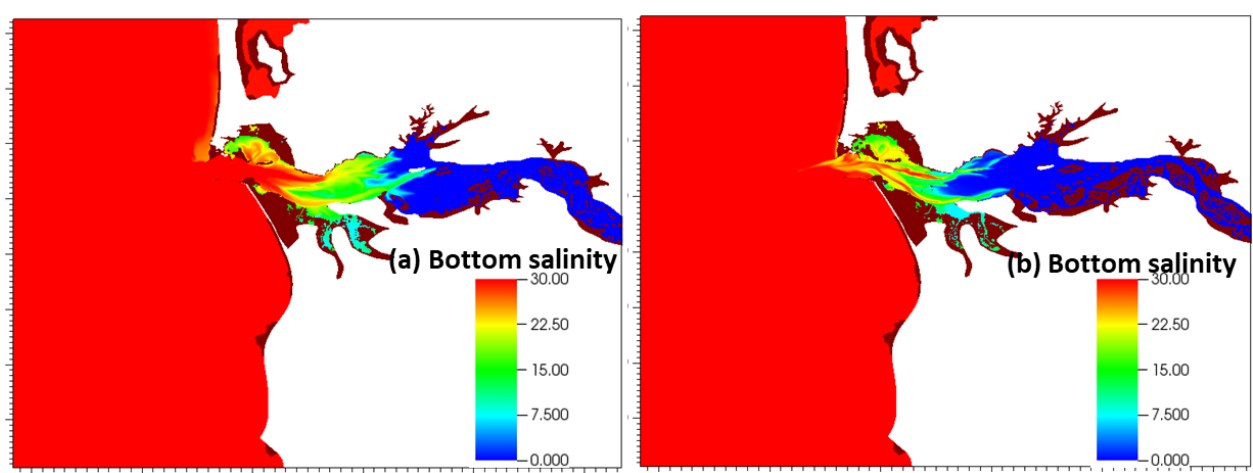

**Fig. 15: Salinity intrusion during (a) flood; (b) ebb in a spring tide. The burgundy areas are dry.**




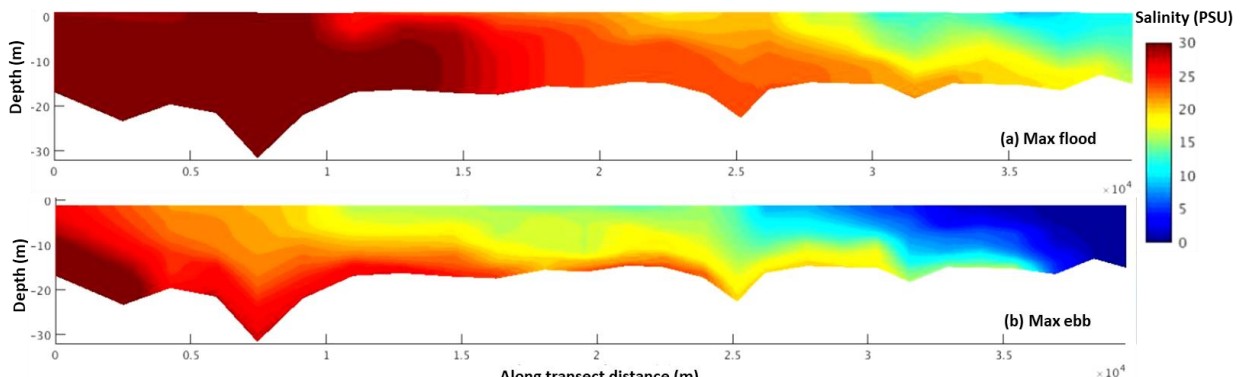

**Fig. 16: Salt intrusion near the peak of (a) flood and (b) edd during a neap tide, along the south channel of the Columbia River (see Fig. 2b for the transect location).**

## 5. Conclusion

We have developed a new 3D unstructured-grid (UG) model (SCHISM) for simulating the global ocean together with coastal ocean and even estuaries in a single mesh, with high resolution applied in the latter. The simulated total elevation (including both tidal and non-tidal components), temperature and salinity have been validated against satellite and in-situ observation data. The simulated tide showed good skill, with a mean complex RMSE of 4.2cm for M2 and 5.4cm for the 5 other major frequencies in deeper depths. The non-tidal residual assessed by the global tide gauge dataset (GESLA) had a mean RMSE of 7cm. The mean MAE for SST excluding Arctic is ~0.8ºC. Larger errors were found for SST in the high latitudes, mostly due to the absence of an ice model.

For the first time ever, we demonstrated the potential for seamless simulation, without the need for grid nesting, from the global ocean into a few estuaries in the US west coast in very high resolution. The model was able to accurately capture the total elevation, and qualitatively capture the challenging salinity intrusion processes in the Columbia River.

Even though the 3D model is more expensive than 2D models, the improved accuracy and ease of calibration for the total water levels justify its cost; this advantage is in addition to the obvious benefits of being able to predict other 3D variables (velocity, temperature and salinity etc). With adequate computational resources, the model can effectively serve as the engine of a global tide-surge and even compound flooding forecasting framework. More meshing work and calibration will be required to further improve its accuracy in specific regions and the UG nature of the model greatly simplifies the required work. In addition, a global tidal model would greatly benefit from transitioning from a tidal datum based to geoid based model to allow more accurate simulation for the total elevations.



### Author contribution

All authors contributed to writing of the paper. YJZ and WP worked on the model setup. TF-M did analyses of the GESLA and altimetry. HY and LC did the analyses for satellite SST and ARGO. SM advised and managed the project.

### Competing interests

The authors declare that they have no conflict of interest.

### Acknowledgement

This study was partially supported by NOAA Water Initiative (NA20NOS4200205). WP was supported by the U.S. National Oceanic and Atmospheric Administration (NOAA) through a Virginia Institute of Marine Science subaward under a Strategic Partnership Project agreement A21162 to Argonne National Laboratory through U.S. Department of Energy contract DE-AC02-06CH11357. Simulations used in this paper were conducted using the following computational facilities:

(1) William & Mary Research Computing for providing computational resources and/or technical support (URL: https://www.wm.edu/it/rc)

(2) the Extreme Science and Engineering Discovery Environment (XSEDE), which is supported by National Science Foundation grant number OCI-1053575;

(3) Texas Advanced Computing Center (TACC; Grant EAR21010), The University of Texas at Austin.

The ARGO data used are from the ARGO project (https://argo.ucsd.edu/data/acknowledging-argo/).

### Code availability

Source code of SCHISM can be accessed at https://zenodo.org/record/6537527.

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
