# Peer review of "Global seamless tidal simulation using a 3D unstructured-grid model (SCHISM v5.10.0)"

_Geoscientific Model Development, 2022_

## Author Response (AR1)

*Our response is shown in red italic texts below. The page and line numbers refer to those in the track-change version.*

**Response to Reviewer 1**

In this paper, authors present a new 3D unstructured-grid global ocean model to study both tidal and non-tidal processes, with a focus on the total water elevation. Unlike existing global ocean models, the new model resolves estuaries and rivers down to ~8m without the need for grid nesting. The model is validated with both satellite and in-situ observations for elevation, temperature and salinity. The authors demonstrate the potential for seamless simulation, on a single mesh, from the global ocean into several estuaries along the US west coast. The model is able to accurately capture the total elevation, and qualitatively capture the challenging salinity intrusion processes in the Columbia River. The model May potentially serve as the backbone in a global tide-surge and compound flooding forecasting framework. There is no doubt about the importance, innovation and value of this work. However, in the writing and presentation, some places still need further explanation. I think this paper satisfies the scope and standard of Geoscientific Model Development, but I also have some concerns. Therefore, I recommend a major revision.

*>> Thank you so much for your constructive comments.*

Major concerns:

- The usual numerical simulation of multi-constituents of tides basically considers four major diurnal and semi-diurnal tides, or eight major diurnal and semi-diurnal tides. Six of the eight main sub-tides are considered. In fact, Q1 is weaker than K2 and P1 tides. The author considers Q1 tides in the study, but does not consider K2 and P1 tides, which gives a strange feeling. Of course, the authors may have made this trade-off from a computational standpoint. The reviewer still insisted that either four major diurnal and semi-diurnal sub-tides or eight major diurnal and semi-diurnal sub-tides should be considered. In view of the fact that the author did not display and analyze the results of N2 and Q1 tides, it is suggested that the author remove these two tides.

*>> We have actually tested including 8 constituents (with additional K2 and P1) but now decide to follow Pringle et al. to include only 5 constituents (by dropping Q1) in the text to facilitate the comparison with previous works (see the discussions after Eq. (5)). Fig. 3f shows the results using 8 constituents. The impact on the M2 complex error is negligible among all cases. Note that we did include the N2 error statistics shown after Eq. (5), to be consistent with Pringle et al. Also, adding more constituents has minimal impact on the computational efficiency.*

- In Page 3, Line 89-91, These features have allowed a single model to be used for challenging compound flooding studies that involve coastal transition zones between hydrodynamic and hydrologic regimes, forced by **ocean, precipitation and watershed rivers** (Ye et al. 2020; Zhang et al. 2020; Huang et al. 2021; Ye et al. 2021). Necessary modifications are required. Where ocean, precipitation and watershed rivers are not a juxtaposition.

*>> We agree and have added a sentence near line 100: "Global compound flooding processes are not the focus of this paper."*

- 3.In the reviewer's opinion, it is very difficult to understand that the bottom friction coefficient is set to 0 in deep water. Please confirm and give a clear explanation. The following is the

corresponding expression in the original text, the reviewer really difficult to understand the bold part of the expression.

Page 4, Line 110, As a result, the near-bottom vertical layers can be as thick as 1km in the deep ocean; in other words, the logarithmic layer there is not well resolved and therefore, we apply **zero friction in the deep depths**.

Page 4, Line 111-113, To ensure adequate energy dissipation toward shallows, we use a simple depth-dependent bottom friction coefficient (used in the quadratic drag formulation) that linearly increases **from 0 at depth 200m to 0.0025 at 50m**.

*>> We have revised the texts (cf. near lines122-128) in view of your comments and also Reviewer #2's. To save computational cost the vertical grid coarsely resolves the deeper depths such that some bottom layer has thickness of ~1km. Therefore, the assumption of a logarithmic velocity profile in the bottom boundary layer may not be appropriate. We have also tested with a very small drag coefficient (1.e-4) in lieu of 0 and the results are quite similar.*

- Page 4, Line 106-107, The number of sigma layers varies from a maximum of 34 to 1 (i.e. 2DH configuration), with **an average of 32 layers**. What does average mean? How it's calculated?

*>> The $LSC^2$ vertical grid system in SCHISM allows different numbers of layers to be applied at different horizontal locations, and the average number of layers is simply the arithmetic mean of the number of layers at each node. We have added an explanation near line 117.*

- In Page 5, Line 138-139, Relaxation of temperature and salinity near the ocean surface, which is commonly utilized in many global ocean models (Ringler et al. 2013), was not applied here due to the relatively short duration of the simulation. This is an awkward statement. What does the author mean by this expression? Ask the authors to give an explanation.

*>> In climate models it's a common practice to relax near-surface T,S to climatological values in order to prevent drift of simulation over a long time. This is not used here. We've added some explanation in the manuscript.*

- In Page 8, Line 196-199, The averaged complex RMSE for M2 is 4.2cm for depths greater than 1km, and 14.3cm for shallower depths. The averaged RMSE for the remaining frequencies (S2, N2, K1, O1, Q1) is 5.4cm / 16.6cm or depths greater/less than 1km. These results are slightly better than the previous best 3D model results without data assimilation (Schindelegger et al. 2018) but worse than those in Pringle et al. (2021). Here the reviewer thinks it must be pointed out that in the numerical simulation of multi-tidal, the evaluation indexes of all sub-tidal should be given in detail. In this paper, the author gives the index of M2 sub-tide alone, and the other five sub-tide indicators are combined.

*>> We have now added those numbers in Abstract and after Eq. (5) (note that we have dropped Q1). We have also replaced the RMSE for the other 4 constituents with the total RMSE from all constituents and compared that with Pringle et al. because the total RMSE is a better metric.*

- Page 8, Line 203-206, Compared to other global 3D models, our model seems to be able to obtain satisfactory results without the need for some elaborate drag formulations described in A18, which might be attributed to the fact that the higher resolution used in the coastal ocean has provided adequate energy dissipation. I don't agree with the author here, the higher resolution used in the coastal ocean is not a panacea.  Ask the authors to give an explanation.

*>> We have rephrased this as it's speculative (near line 234). Higher resolution might be one reason but there may be others.*

- Page 10, Line 245, Table 1. Summary of model performance to reproduce the main semi-diurnal tidal component for SCHISM and FES2012 models against GESLA. Are the indicators here consistent with those in lines 196-199? I think this table currently lacks a clear interpretation.

*>> The GESLA comparison is complementary to the co-tidal chart because it's focused on tide gauges. We used the same Eqs. (3-4) to compute the error metrics; since GESLA observations are at point locations, simple averaging instead of area averaging is done as in Eq. (5). We have added some explanations in the text and table caption.*

Minor concerns:
Abstract, Line 14-15, Tidal elevation solutions have a mean complex RMSE of 4.2 cm for M2 and 5.4 cm for combined 5 other major frequencies in the deep ocean. I think the evaluation indexes of all sub-tidal should be given in detail. Where frequencies should be constituents.

*>> Added (and dropped Q1).*

Page 8, Line 197ï¼Œthe remaining frequencies should be the remaining constituents

*>> We have changed 'frequency' to 'constituent' here and elsewhere.*

Page 16, Line 326, systems should be systems

*>> Corrected.*

Page 17, Line 356, The large stratification should be The strong stratification

*>> We have removed that paragraph.*

Page 20, Line 391-392, The simulated tide showed good skill, with a mean complex RMSE of 4.2cm for M2 and 5.4cm for the 5 other major frequencies in deeper depths. What is deeper depths mean?

*>> Changed to 'in depths greater than 1km'.*

**Response to Reviewer 2**

This manuscript presents a 3D global ocean set-up within the SCHISM model and demonstrates the ability of the model to hindcast some aspects of the sea surface level, salinity and temperature distributions. As a main feature, the authors highlight the seamless (i.e. within a single grid) integration of global oceans and high resolution coastal features and estuaries in a 3D baroclinic model, which is new.

The modelling framework presented by the authors is impressive and results on SSH, salinity and temperature on the global level seem state-of-the-art, going by the numbers provided by the authors themselves. To me however, the manuscript reads like a progress report for the community already using SCHISM, not like a GMD paper. For it to become a GMD paper, I expect the authors to identify the unique ability of their model and then show it actually delivers on this ability. The authors identify the cross-scale feature as unique, but do not show this aspect actually delivers anything and why one would want this feature. Also, while results on the estuary scale are presented, it is unclear how accurate they are and hence unclear whether the model is usable for the estuary scale. I detail my comments below.

Overall I recommend major revisions.

*>> Thank you so much for your constructive suggestions. We have revised the manuscript to address your concerns and comments. As you mentioned, the main novelty of the paper is the seamless simulation from global to local scales in a single model for the total surface elevation. We believe we have now delivered this promise with the simulated elevations inside two estuaries in US west coast, including a very challenging upstream station (Fig. 13d) that was added at your suggestion. We believe this unique capability is of wide interest.*

**Main comments**

**1. Context and aim**

The context now set in the introduction is that coastal and estuarine areas and baroclinic (3D) processes are of added value for evaluating the global ocean energy budget properly and that this SCHISM set up provides a new and unique environment for including these aspects. However, nowhere in the manuscript the added value of shallow areas or baroclinic processes to the global SSH or energy budget are actually shown, let alone that it is shown that this model is therefore superior or adds new understanding. In several places (ln 201-202, 303-306) it is claimed that the baroclinic processes simplify calibration and improve performance, but no evidence of this is given. The authors only compare their model on global M2 tides, SST and SSS but the added value of the shallows and baroclinicity to these results is not shown.

This is a GMD paper, so no new scientific results are needed here, but I do expect the authors to say why I would want the unique ability of their model and then show that they can actually deliver that. Hence, if the context is, for example, the global energy budget, please compute the energy budget of the shallow areas and of the baroclinic component and compare to literature. The authors may respond to this comment by changing the context in the introduction and/or adding new results. Essential is that they demonstrate the model results can deliver on the sketched context and the specific novelty of this model.

*>> To illustrate the importance of shallows, we have added a plot (Fig. 3e) from a simulation with <50m treated as dry land. We referred to Blake et al. (2022) who showed the need for elaborate calibration for 2D model. This is also consistent with our own experience with our own SCHISM 2D model (Fig. B1 below). The comparisons with 2D models demonstrate the importance of baroclinicity. Most importantly,*

*the new model simulates the total water levels (tides, surges and eventually compound inland-coastal flooding, although the last is not demonstrated in the paper) without the need for open boundary conditions (as in the case of regional models), all the way into up estuary (Fig. 13d). It therefore can fill a critical gap in forecasting. We have revised the Intro to reflect these points and the main novelty of this study (cf. near lines 58-66).*

[Figure]

*Fig. B1: Comparison of M2 co-tidal chart from (a) 3D model; (b) 2D model (after extensive calibration using spatially variable bottom friction). Note that several amphidromic locations are misplaced in the 2D results (e.g., in the Southern Ocean and eastern Pacific). (c) TPXOv9 reference solution.*

**2. Focus on certain processes and mentioning of other global ocean models**

The manuscript to me hanging between two conflicting ideas. On the one hand, the authors state they want to focus on M2 tides and the addition of shallows, while they accept less focus on some other processes (e.g. eddies, sea ice). On the other hand, I did get the feeling from ln 37-43 and the manuscript overall that the authors are aiming at improving global forecasts of tides, SST and SSS. This is followed by a great deal of apologies for barely resolving eddies (e.g. Ln 173, 235) and sea ice (154, 257, 272, 278, sect 4.1) and no focus on any improvement compared to other models. These apologies are not so nice to read and feel disappointing. I'd advice the authors to clearly state their assumptions and then briefly mention several global models (with names and references) and some of their unique abilities in terms of resolution, dimension, eddies, sea ice etc. Then they can follow on the first idea of focusing on shallows only and accepting their assumptions. The missing eddies and sea ice then only need to be mentioned briefly to explain some of the discrepancies in the results, but no more is needed. Results on the specific

added value of shallows and baroclinic processes is needed. Such an overview of models and their abilities helps readers to set the results in context and select the model that they need for their purpose.

*>> The focus of the paper is on the total elevation (not just M2 tide) in both deep and shallow, and since the baroclinicity is known to be important for that (including extreme levels; Calafat et al. 2018), we chose to validate SSS and SST etc. We have revised the Intro to sharpen the focus on shallows and total elevation, and summarized review of other global models in the new Table 1. We have also added (1) a comparison with SCHISM 2D model results to show the importance of baroclinicity (Fig. B1); (2) results from the 3D model with areas shallower than 50m cut off to show the importance of shallows (Fig. 3e); (3) a challenging upstream station in San Francisco Bay to demonstrate the seamless cross-scale capability (Fig. 13d). Redundant descriptions/apologies have been removed and model caveats have been centralized near the end of Section 2.2 (paragraph starting from line 173).*

**3. Estuaries**

Section 4.2 about estuaries should play an essential role in this manuscript, as it is the main section that focuses on the cross-scale feature that is the main novelty of this model. However, I don't know what to learn from this section. It is shown that some degree of stratification and region-of-fresh-water-influence (ROFI) is produced by the model. However, since there is no systematic comparison of salinity with observations or dedicated models, I have no idea if results are accurate and for what purpose these estuary-scale results could be used. As the authors state themselves, more local calibration will be needed in estuaries and this is not efficient to do in a global model. So then why would an estuarine scientist want this? I don't think the authors can state with any confidence that this global model can be reliably and practically (i.e. time-wise) be used to say anything about estuaries. Hence, I think this section needs to be removed, possibly to consider for follow-up research. Instead, the authors may choose to focus on water levels only. In this case, results and observations further up-estuary would be appreciated and I would like to know why I want this over a local model.

In their own introduction, the authors never claim the model is suitable on the estuary level. They only claim that the addition of estuaries has important back effects on the global SSH and energy budget. In line with my comments above, I therefore recommend the authors to focus on this aspect instead.

*>> We agree and have revised Intro and Section 4.2 (Section 4.3 in the revised version) to focus on the total water levels (and added a challenging up-estuary station in SF Bay). The comparison with the simulation with shallows removed serves to highlight the importance of shallows (Fig. 3e). Still, we think that even a qualitative skill on Columbia River plume is a step forward as no other global models were able to achieve this even at a qualitative level. Also, there are numerous process-based studies on plumes that sought to understand the physics without delving into quantitative comparisons (Garcia-Berdeal et al.2002; Fong and Geyer 2002). Therefore, we have retained a short discussion near the end to reflect this and as an introduction into future work. Even though calibration of the global 3D model in estuaries is relatively expensive, we believe demonstrating this new potential (with a few exemplary estuaries) is novel.*

**4. Context of SCHISM**

SCHISM is not new and the authors add several references to the model. It remained unclear to me if the model presented here adds any new aspects to the software or if this is a new global grid set-up within the existing software. I recommend the authors to add a short subsection stating what is actually new about this model and the set-up and list a few cases where the model was used previously, also to demonstrate the novelty of this manuscript.

*>> We have added a paragraph to explicitly mention the new development made in this work (near line 150).*

**5. Reproducibility**

Please add the version number/date of SCHISM used. Is the grid setup and forcing data all available online?

*>> The version has been added in the title and Code Availability section. The model setup has been uploaded to our own web site: <http://ccrm.vims.edu/yinglong/TMP/Glb/RUN06a/> due to its large size.*

**Specific comments**

Ln 97: salt intrusion processes? (1) this is new and was not mentioned in the introduction at all. Given the goals, it seems unfit to discuss this. 2) even models focussed on estuaries alone struggle to properly resolve salt intrusion dynamically, so even if you have the right resolution does not mean you can actually resolve it well.

*>> Removed.*

Ln 107: where are the small numbers of layers used. Baroclinic effects in shallow areas can be important for friction estimates and hence the local energy budget. Also, it seems you don't use such few layers in all shallow areas. A bit more elaboration on your choices would be useful here.

*>> Actually, we used adequate number of layers for the salt intrusion study. The first master grid has 1 layer at depth of 0.4m or shallower. The 2$^{nd}$ master grid has 23 layers at depth of 10m so the baroclinic effects should be well resolved. We have added this in the text (near line 120).*

Ln 111: no friction at large depth is probably a reasonable approximation for many purposes, but is this not a huge error compared to the energy dissipation in an estuary? When reading this I'm still on the line that you want to improve global energy budgets. In that context it seems weird to model details of estuaries but be so crude about friction in the global ocean. A much improved context in the introduction should help to resolve this.

*>> We note that most other 3D global models also use simple bottom friction schemes. We have revised those sentences. Note that the bottom friction is only one component of dissipation; others include internal tides and turbulence etc that have been included in the model. Furthermore, using a small friction coefficient (1.e-4) gave similar results.*

Internal tides (e.g. Ln 116): is its dissipation in the model not because of reduction of number of layers towards the shallow? Did you check numerical accuracy of any of the baroclinic action in the global model?

*>> Validation of ITs is on-going and out of scope for this paper, although we do have some evidence in some localized areas (e.g. near Taiwan). As we used 23+ layers for depths >=10m, IT dissipation should be reasonably represented in the model.*

Ln 201-
202 'the more complete physics ...' This conclusion is not supported at all by results. Please remove.

Again in ln 303-306. The more complete physics does not generally simplify calibration. I believe you have a specific benchmark model in mind that includes parameters for the baroclinic processes that you don't need. Also, you choose not to add any additional calibration parameters related to the baroclinic part (you could choose to add some parameters to the turbulence closure for internal wave breaking and such). Please make this explicit. What benchmark model(s) are you comparing to? What specific parameters do you not have (or are simpler compared to that/those models)?

*>> Fig. B1 above shows comparison with our own 2D model, which has been extensively calibrated. In the case of the 3D model, as you can see from the paper, we used standard turbulence closure scheme and a simple viscosity scheme; no effort was made on calibration except for the bottom friction (which is a simple function of the depths). In the case of our 2D model, the results are still not as good even after extensive calibration on the bottom friction. We have added some explanations near line 230. In short, even though it may sound counter-intuitive as the 3D model is more expensive and involves more parameters than the 2D model, the calibration process for elevation is actually much easier (it's a different story for other 3D variables). This has been confirmed with other basin-scale studies (Huang et al. 2021, 2022; Ye et al. 2020) and was behind the motivation for another 2D model to add more physics (Pringle et al. 2019, 2021).*

Ln 270 and fig 8: indeed, river plumes seem to be hugely overestimated. Based on this, I'd say that your model cannot be used to look closer at any coastal areas or estuaries specifically (hence, section 4.2 should be removed). Also, I'm not sure if your addition of shallow areas is actually having the correct back-effects on the global SSH or energy budget. Substantiation of this in numbers and showing these effects of coastal areas on the global values are appropriate is really needed.

*>> We have added a new Section 4.2, and a comparison with shallows removed (Fig. 3e) to demonstrate the importance of shallows. The exaggeration of plumes may be rectified by following a similar procedure as we did for the Columbia River estuary; a main constraint is the lack of quality DEMs in some regions.*

Ln 344-347: you claim here that your model better captures the NTR compared to 2D models. However, I have not seen the NTR in fig 13 and it is not compared to 2D models. This claim is unsupported. I think it should be removed.

*>> We have removed 'over 2D models'.*

Fig 14-16 look nice, but do not substantiate anything if there is no comparison to observations and no indication in the text on how one should want to use this model for estuary-scales.

*>> We think that even a qualitative skill on Columbia River plume is a step forward as no other global models were able to achieve this even at a qualitative level. Also there are numerous process-based studies on plumes that sought to understand the physics without delving into quantitative comparisons. Our own experiences strongly suggest that a qualitative agreement is often the more challenging part of calibration exercise and bodes well for the next step of quantitative calibration. Therefore, we have retained some discussions and Fig. 14 near the end but removed Figs. 15-16, as a way to introduce our future work.*

Ln 365: adequate: what is that? I don't see comparison to observations or other dedicated models

*>> Removed.*

Ln 367-370: this is speculation without substantiation. I don't think this fits here.

 *>> Removed.*

Eq 2 seems to have a printing error

*>>Corrected.*

Ln 148: What is SAL?

*>> It stands for self-attracting and loading (defined near line 150).*

Eq 5: integral should be over Omega not A. This is a form of the average, not integrated (ln 192)

 *>> Corrected.*

Ln 332: edge -> wedge
 *>> Corrected.*

**References**

Calafat, F.M., Wahl, T., Lindsten, F. et al. (2018) Coherent modulation of the sea-level annual cycle in the United States by Atlantic Rossby waves. Nat Commun 9, 2571. https://doi.org/10.1038/s41467-018-04898-y

Fong, Derek & Geyer, W. (2002). The Alongshore Transport of Freshwater in a Surface-Trapped River Plume. Journal of Physical Oceanography - J PHYS OCEANOGR. 32. 957-972. 10.1175/1520-0485(2002)032<0957:TATOFI>2.0.CO;2.

---

## Author Response (AR2)

*Our response is shown in red italic texts below. The page and line numbers refer to those in the track-change version.*

**Response to Reviewer 1**

The paper has made some progress after the authors revised it. For the questions raised by reviewers, some answers are satisfactory, but there are other parts that the author needs to continue to answer and give convincing evidence.

1.In the reviewer's opinion, it is still very difficult to understand that the bottom friction coefficient is set to 0 in deep water. The following is the corresponding expression in the revision text, the reviewer still really difficult to understand the bold part of the expression.

The vertical high resolution is focused on the near-surface zone at the expense of the bottom in order to conserve computational cost. As a result, the near-bottom vertical layers can be as thick as 1km in the deep ocean; in other words, the logarithmic bottom boundary layer at deep depths is not well resolved and therefore, we apply zero friction in the deep depths. Alternatively, using a small friction coefficient (10-4) gave similar results. To ensure adequate energy dissipation toward shallows, we use a simple depthdependent bottom friction coefficient (used in the quadratic drag formulation) that linearly increases from 0 at depth 200m to 0.0025 at 50m (i.e., 0 friction is used at depths deeper than 200m and 0.0025 is used at depths shallower than 50m, with a linear transition in between the two depths). For the sake of simplicity, no attempt has been made to optimize the friction in each region yet, and this is left for future work.

I don't think the author has fully explained it clearly here. There are still some questions: Are the authors clear about the specific expression of friction coefficient and its physical meaning? According to the authors, the bottom friction coefficient is 0.0025 when the water depth is less than 50 meters, while the bottom friction coefficient is 0 or 0.0001 when the water depth is more than 200 meters, which is very inconsistent with our understanding of the bottom friction coefficient in the tidal model. The author gives the impression here that the bottom friction coefficient is assigned different values at different depths in a water column. The reviewer really think it is difficult to understand. Please confirm and give convincing evidences.

*Sorry for the confusion about 1.e-4. We have revised the texts on pg 5. Basically, we specify the drag coefficient $C_d$ based on the local depth:*

$$C_d = max\{C_{d2}, min[C_{d1}, C_{d1} + (C_{d2} - C_{d1}) * (h - h_1)/(h_2 - h_1)]\}$$

*where $h$ is the local depth, $h_1 = 50m$ and $h_2 = 200m$ are the two transition depths with corresponding friction coefficients of $C_{d1} = 0.0025$ and $C_{d2}$ respectively. In the baseline setup we used $C_{d2} = 0$, but we have also tried $C_{d2} = 0.0001$. There is a unique value for $C_d$ along each water column.*

*So, we do in fact use a 0 or 0.0001 friction coefficient in deep water, but this is justified by the large thickness of the bottom layer in deep water (~1 km). This doesn't mean we think that the actual boundary layer in deep water is frictionless; this is more a numerical treatment.*

2.In the text, The averaged complex RMSE for M2 is 4.2cm for depths greater than 1km, and 14.3cm for shallower depths. The averaged total RMSE for all constituents) is 5.4cm / 16.6cm or depths greater/less

than 1km. The breakdown of RMSEs for the other 4 constituents (S2, N2, K1, O1) is: 2.05cm, 0.93cm, 2.08cm, and 1.34cm for depths greater than 1km; 6.07cm, 2.60cm, 4.71cm, and 2.84cm for depths shallower than 1km. These results are slightly better than the previous best 3D model results without data assimilation (Schindelegger et al. 2018) but slightly worse than those in Pringle et al. (2021); e.g., the total RMSE from their model is 3.9 cm / 17.2 cm in the deep/shallow ocean respectively. But in Table 2, RMSE M2=32cm, RMSE S2=11cm. What's wrong with the inconsistency between the above two? I feel that the results in Table 2 may be problematic.

The inconsistency in RMSE is because the comparison was done against different datasets: TPXOv9 or GESLA. The latter consist of world-wide tide gauges, many of which are located in sheltered coastal areas that have larger uncertainties in DEMs used or have not been resolved yet in the current mesh. The GESLA comparison is therefore inherently much more challenging. As you can see from Table 2, even FES (which assimilates tide gauges) has larger errors. We have added some explanation on pg 11.

3. In Table 2, unit is missing. What does the formula (5) have to do with Table 2?

About Formula (5), how do you calculate the area of the corresponding part?

Units are now added in Table 2. We are not sure about your question on Eq. (5) and Table 2; they are not related. The numbers shown in Table 2 are calculated from Eq. (6), which uses Eq. (3). Note that the averaging here is done over all tide gauges, not over areas.

The area in Eq. (5) is the union of all mesh elements. The RMSEs are first calculated in each element before area-integration.

4.The color bar in Figure 3 should be corrected. In Figure 3, I think the differences of a and b, a and d, a and e, a and f, may be a good choice. Where a stands SCHISM3D, b stands TPXOv9, d stands Antarctica Shelf, e stands Shallow removed, and f stands 8 constituents.

We have added color bars for some subplots. The differences between (a) and (b) is (c). The differences for some other subplots are tricky to compute. Computing complex differences between (a) and (e) is problematic because the area in (e) is reduced from (a). In any case, the qualitative differences as described in the text are much more telling than the quantitative differences.

**Response to Reviewer 2**

I thank the authors for their revisions. My previous comments have been addressed. Specifically, I think it is more clear that the model is not just similarly good at simulation of tides as other models, but does so with less calibration effort. Proof of this calibration effort relies somewhat on experience of the authors, but I think it is now sufficiently motivated. I'm still not convinced that the coastal and estuarine small scale processes beyond tides are sufficiently well resolved/calibrated so that they can be studied in isolation in this model, but the authors also discuss this properly. I think therefore the demonstration of the estuarine scale is now appropriate.

Thank you for your support.

I have some minor suggestions and otherwise am happy to accept this manuscript for publication.

Comments

- The link to the 3D model set-up is now included under 'code-availability'. This refers to a VIMS website. I'm not sure what the rules of the journal are with respect to code repositories and I recommend the authors (or editorial team) to check if this website satisfies the criteria.

We have now uploaded the zipped files to Zenodo with a DOI (see Code Availability section). The model input file size (127GB) is large so we had to split it into 4 pieces.

- Near line 150 (or elsewhere): could you refer to documentation of SCHISM, to refer readers to all features of the model?

We have added a sentence on pg 6 to refer readers to continuously updated online manual on SCHISM web site (schism.wiki) that explains details of all features.

- Ln 150-153: in point 2) you claim that your use of 'scribe' cores significantly improves parrallelisation and scaling. If available, please refer to documentation that shows this.

The details are explained in the continuously updated online manual. In addition, this claim has been corroborated by many community users.

- Ln 360: you cannot technically conclude that your results are consistent with 70-75% dissipation of energy on shallows as you did not show an analysis of the energy budget. Please weaken the statement that it is consistent with earlier findings that dissipation on shallows is important for the global tidal amplitude.

Revised on pg 18.